# Micro Ribonucleic Acid−29a (miR−29a) Antagonist Normalizes Bone Metabolism in Osteogenesis Imperfecta (OI) Mice Model

**DOI:** 10.3390/biomedicines11020465

**Published:** 2023-02-05

**Authors:** Jih-Yang Ko, Feng-Sheng Wang, Sung-Hsiung Chen, Shu-Jui Kuo

**Affiliations:** 1Department of Orthopedic Surgery, College of Medicine, Chang Gung University, Kaohsiung Chang Gung Memorial Hospital, Kaohsiung 833401, Taiwan; 2Center for Shockwave Medicine and Tissue Engineering, College of Medicine, Chang Gung University, Kaohsiung Chang Gung Memorial Hospital, Kaohsiung 833401, Taiwan; 3Department of Medical Research, College of Medicine, Chang Gung University, Kaohsiung Chang Gung Memorial Hospital, Kaohsiung 833401, Taiwan; 4School of Medicine, China Medical University, Taichung 404328, Taiwan; 5Department of Orthopedic Surgery, China Medical University Hospital, Taichung 404327, Taiwan

**Keywords:** osteogenesis imperfecta, microRNA-29a (miR-29a), proliferating cell nuclear antigen (PCNA), dickkopf-1 (DKK1), β-catenin

## Abstract

Osteogenesis imperfecta (OI) is not curative nowadays. This study tried to unriddle the therapeutic potential of micro ribonucleic acid-29a (miR-29a) antagonist in treating OI in a mouse animal model (B6C3Fe a/a-Col1a2oim/J). We showed that the expression levels of miR-29a were higher in bone tissues obtained from the OI mice than from wild-type mice demonstrated by reverse transcription-quantitative polymerase chain reaction (RT-qPCR) and in situ hybridization assay. We established lentivirus-shuttled vector expressing miR-29a antisense oligonucleotide (miR-29a-AS) and miR-29a precursors (pre-miR-29a), showing that the inferior bony architecture in micro-computed tomography and pertinent morphometric parameters could be rescued by miR-29a-AS and deteriorated by pre-miR-29a. The decreased proliferating cell nuclear antigen (PCNA), increased Dickkopf-1 (DKK1), and decreased β-catenin expression in OI mice could be accentuated by pre-miR-29a and normalized by miR-29a-AS. The decreased osteogenesis and increased osteoclastogenesis in OI mice could also be accentuated by pre-miR-29a and normalized by miR-29a-AS. miR-29a-AS did not seem to possess severe hepatic or renal toxicities.

## 1. Introduction

Osteogenesis imperfecta (OI) is a group of autosomal dominant genetic diseases associated with pathogenic variants of the COL1A1 and COL1A2 genes [1,2]. COL1A1 and COL1A2 encode the alpha chains of collagen type 1, the major constituting component of bones [3,4]. The aberrant expression of COL1A1 and COL1A2 could contribute to defects in bone formation, and OI patients could present with bone deformities, compromised bone strength, and other connective tissue presentations such as soft teeth scleral discoloration [1]. The revised Nosology and Classification of Genetic Skeletal Disorders defines 5 forms of OI: persistent blue sclera without deformity (OI type I), lethality at birth (OI type II), progressive deformity (OI type III), moderate severity (OI type IV), and with hypertrophic callus and/or calcification of the interosseous membranes (OI type V) [5]. Type I OI has the least severity, while patients with type III OI are most gravely affected among the surviving OI patients, suffering from short stature, frequent fractures, and restricted mobility [1].

Currently, there is no eradicative therapy for OI. Adequate calcium and vitamin D intake could just refine bone mineralization [6,7]. Bisphosphonates could increase lumbar spine bone mineral density and reduce the fracture risk among children with severe OI [6,7]. Denosumab improved lumbar spine bone mineral density (BMD) in a small patient cohort [6,8]. The regimens mentioned above are not free from side effects, and the safety profile of life-long application of these agents are not available at present. These clinically unmet need urges our team to search of alternative treatments for OI.

Micro ribonucleic acids (miRNAs) are the noncoding short single-stranded ribonucleic acid (RNA) with 18–24 nucleotides mediating the silencing of target gens in the post-transcriptional level [9,10]. Recent studies have elaborated many facts suggesting the close relationship of miRNAs in the processes of bone formation, growth, and remodeling [11]. For example, miR-34, miR-182, miR-199, miR-214, and miR-2861 are associated with osteoblast functions during postnatal skeletal development [12]. MiR-29b, miR-125b, miR-204, miR-211, miR-133, miR-135, miR-378, miR-2861, and miR-3960 are involved in the process of osteoblastic differentiation [12]. Moreover, the dysregulation of miRNA expression is correlated with the development of rheumatoid arthritis (e.g., miR-16, miR-124a, and miR-155); osteoarthritis (e.g., miR-34a, miR-181a, and miR-455-3p); and osteoporosis (e.g., miR-133a, miR-138, and miR-214) [12]. Considering the facts mentioned above, miRNAs are being considered as a new generation of therapeutics for bone diseases [11]. Moreover, microRNAs have also been shown to be involved in the pathogenesis of OI [13,14], and serum microRNAs have been suggested as a promising biomarker for OI [15]. Due to the emerging interests in the application of miRNAs for the treatment of bone diseases and the known linkage of miRNAs and OI pathogenesis, we aim to unravel the therapeutic potential of miRNAs for the treatment of OI.

Our team has been dedicated in unravelling the role of miR-29a in various bone diseases, including bone loss secondary to glucocorticoid, estrogen loss, and high fat diet in recent years [16,17,18,19,20,21]. In this study, we want to unravel the role of miR-29a in the bone metabolism of OI animal model. We hypothesize that miR-29a harbors the normalizing potential for bone metabolism in OA animal model.

## 2. Materials and Methods

### 2.1. The Establishment of OI Mice Animal Model

The OI mice (B6C3Fe a/a-Col1a2^oim^/J) were purchased from Jackson Laboratory (Bar Harbor, ME, USA) (Strain #:001815, RRID:IMSR_JAX:001815), and this strain has been successfully applied in other study [22]. Homozygous mice exhibit osteopenia, progressive skeletal deformities, fractures, cortical thinning, and small body size. These mice carry collagen 1α2 gene mutant that displayed low bone mineral density and osteogenesis imperfecta phenotype in adult mice. The pyrosequencing protocol suggested by the manufacturer (Protocol 14160: Pyrosequencing Assay) was adopted for genotyping (Jackson Laboratory, Bar Harbor, ME, USA). B6 mice for wild-type group were obtained from Biolasco Biotechnology Co. (Nangang Dist., Taipei City, Taiwan). Animal use protocols (No. 2010121415, No. 2015122306) were approved by the Institutional Animal Care and Use Committee of Kaohsiung Chang Gung Memorial Hospital. Mice were raised in the pathogen-free laboratory animal facility in Kaohsiung Chang Gung Memorial Hospital with drinking water and feeding ad libitum in the milieu of temperature 23 ± 2 ℃ and humidity 55 ± 5%.

### 2.2. The Establishment of Lentivirus-Shuttled Vector Expression miR-29a Antisense Oligonucleotides and miR-29a Precursors

Lentivirus-shuttled vector expressing miR-29a antisense oligonucleotide (miR-29a-AS) and miR-29a precursors (pre-miR-29a) were constructed following the methodologies described in our previous publications [16]. miR-29a-AS and pre-miR-29a were constructed via the pPACKF1 FIV Lentivector Packaging Kit following the manufacturer’s instructions (System Biosciences, Palo Alto, CA, USA). Briefly, the gene coding miR-29a precursor and antisense oligonucleotide were inserted into pMIF-cGFP-zeo and pmiR-ZIP short hairpin RNA expression vectors. The constructs were co-transfected with pPACKF1 vector into 293T cells. After amplification, lentivirus was purified by cesium chloride density-gradient ultracentrifugation (Sigma Aldrich, St. Louis, MO, USA). Titers of lentivirus were determined using the plaque-forming method. Each animal was anesthetized and given 0.2 mL lentivirus suspension containing 5 × 10^9^ plaque-forming units/mL via tail vein injection. At 12 weeks after treatment, animals were euthanatized for subsequent studies.

### 2.3. Micro-Computed Tomography (micro-CT) Assay

Micro-computed tomography (micro-CT) of bone tissues and reconstruction of 400 radiographs (9 µm pixel) into transverse and sagittal views of images were carried out employing 1176 Skyscan scanner (Bruker, Billerica, MA, USA) and SKYSCAN^®^ CT-Analysis software. The BMD of the region of interest (ROI) was assessed upon calibration employing hydroxyapatite phosphate (HA) phantom (750 mgHA/cc, Computerized Imaging Reference Systems, Inc., Norfork, VA, USA). Morphometry of ROI, including BMD (mg/cm^3^), trabecular bone volume to total volume fraction (BV/TV) (%), trabecular number (Tb.N) (1/mm), trabecular thickness (Tb.Th) (mm), trabecular separation (Tb.Sp) (1/mm), and structure model index (SMI) were calculated using the software [21].

### 2.4. Reverse Transcription-Quantitative Polymerase Chain Reaction (RT-qPCR) Assay

Bone tissues were harvested in the following processes for subsequent studies. Intact femurs with marrow were crushed and homogenized by Precellys 24 homogenizer (Bertin Technologies, Montigny-le-Bretonneux, France) under RNase-free conditions. Total miRNA was isolated using microRNA isolation kits (BioChain, Newark, CA, USA). The reverse transcription-quantitative polymerase chain reaction (RT-qPCR) assay was employed to assess the transcription levels of target genes, such as miR-29a. Tissues were pulverized under liquid nitrogen without RNase, and the total RNA was extracted by RiboPure RNA purification kit (Thermo Fisher Scientific Waltham, MA, USA). One microgram RNA was reverse transcribed into complementary deoxyribonucleic acid (DNA) employing Step One Real-Time PCR System (Thermo Fisher Scientific, Waltham, MA, USA) complying to the manufacturer’s guidance. The threshold cycle (Ct) was defined as the number of cycles needed for the fluorescence signal to become discernible. The ΔCt and ΔΔCt were defined by the equation.
ΔCt = Ct (target gene mRNA, such as miR-29a) − Ct (18S rRNA)(1)
ΔΔCt = ΔCt(study group) − ΔCt (control group)(2)

The mRNA expression levels of the assayed genes by the study group were expressed as 2^−ΔΔCt^ [23,24].

### 2.5. In Situ Hybridization

In situ hybridization for sections was performed using IsHyb in situ hybridization kits (BioChain, Newark, CA, USA) according to the manufacturer’s instructions. MiR-29a transcripts in sections were probed by digoxigenin (DIG)-labeled miR-29a probes (Exiqon, Shilin Dist., Taipei City, Taiwan), followed by incubation with horseradish peroxidase (HRP)–conjugated anti-DIG antibody for the specimens fixed, decalcified, and embedded under RNase-free conditions [16].

### 2.6. Histologic Analysis

Distal femurs were fixed in 4% phosphate-buffered saline, buffered formaldehyde, decalcified, embedded in paraffin, and then cut longitudinally into 5 μm thick sections. The immunoreactivities of proliferating cell nuclear antigen (PCNA) (Cell Signaling Technology, Danvers, MA, USA), Dickkopf-1 (DKK1) (Santa Cruz Biotechnology, Dallas, TX, USA), and β-catenin (Cell Signaling Technology, Danvers, MA, USA) were detected using respective primary antibodies and non-biotin horseradish peroxidase detection system (BioGenex, San Ramon, CA, USA), followed by counterstaining with hematoxylin, dehydration, and mounting. Those without primary antibodies were enrolled as negative controls for the immunostaining. Ten sections obtained from five mice were measured. Three images from each section were randomly selected, taken, and counted under ×400 magnifications. The number of positive immunolabeled and total cells per high-power field in each section was counted and percentage of positive labeled cells was calculated [25]. Osteoclast distribution in trabecular bone within metaphyseal region was detected by tartrate-resistant acid phosphatase (TRAP) histochemical staining. Ten sections from five mice were randomly selected for microscopy and image analysis. TRAP-stained osteoclast area in each high-power field (×400 magnification) were measured using the Zeiss Image Analysis System (Oberkochen, Baden-Württemberg, Germany).

### 2.7. Assessment of Osteogenic Potential

We assessed the impacts of miR-29a-AS and pre-miR-29a on the osteogenic potentials of the harvested primary bone marrow mesenchymal cells. Briefly, primary bone marrow mesenchymal cells harvested from femurs were mixed with red blood cell lysis buffer (11814389001; Sigma-Aldrich, St. Louis, MO, USA) to isolate mononuclear cells. Upon incubating mononuclear cells in Dulbecco’s modified Eagle medium with 10% fetal bovine serum (FBS) (Thermo Fisher Scientific, Waltham, MA, USA) overnight, adherent cells were collected and incubated in osteogenic medium (10^5^ cells/well, 24-well plates) (StemPro™ Osteogenesis Differentiation Kit; Thermo Fisher Scientific, Waltham, MA, USA) for 18 days. The extent of mineralization was assessed employing von Kossa Stain Kits (ab150687; Abcam, Cambridge, UK), following instructor’s protocol. Calcium in mass deposits would be stained with black color by the Kit, and the black color-stained area (mm^2^/filed) of von Kossa-stained mineralized matrices in each ×125 magnification field were measured applying light microscopy (Zeiss Image Analysis System, Oberkochen, Baden-Württemberg, Germany) [19,21].

### 2.8. Assessment of Osteoclastogenic Potential

Primary bone marrow osteoclast precursors cells were isolated following the protocols described in our previous publications to assess the impact of miR-29a-AS and pre-miR-29a on the osteoclastogenic potentials of the harvested primary bone marrow osteoclast precursors [26]. Briefly, nucleated cells in bone marrow were isolated using RBC Lysis buffer (Sigma-Aldrich, St. Louis, MO, USA) and incubated in α-minimum essential medium (MEM) with 10% FBS and 20 ng/mL macrophage -colony stimulating factor (M-CSF) (R&D Systems, Minneapolis, MN, USA) for 24 h. The floating cells were collected upon incubation. Then, 10^5^/well macrophages (24-well plates) were incubated in osteoclastogenic medium comprising α-MEM, 10% FBS, 20 ng/mL M-CSF and 20 ng/mL receptor activator of nuclear factor kappa-B ligand (RANKL) (R&D Systems, Minneapolis, MN, USA) for one week. F-actin ring formation in osteoclasts was probed employing F-actin antibody conjugated with Alexa Fluor^®^ 488 Phalloidin (Life Technologies, Grand Island, NY, USA) and 4′,6-diamidino-2-phenylindole (DAPI)-Fluoromount G (Southern Biotech, Birmingham, AL, USA). The total number of F-actin rings in 3 different fields (×100 magnification) per well and 3 wells per animals were counted employing a Zeiss inverted microscope and image-analysis software (Zeiss Image Analysis System, Oberkochen, Baden-Württemberg, Germany).

### 2.9. Statistical Analyses

Between-group differences were assessed by the analysis of variance test and Tukey’s HSD (Honestly Significant Difference) post hoc analysis employing GraphPad Prism v5.0 (GraphPad Software Inc., San Diego, CA, USA) [23].

## 3. Results

We employed the pyrosequencing genotyping protocol offered by the manufacturer (https://www.jax.org/Protocol?stockNumber=001815&protocolID=14160, accessed on 1 August 2019). All the adopted OI mice in the subsequent experiments showed a strong band of mutated gene of interest in agarose gel electrophoresis (Figure 1).

Under the reverse transcription-quantitative polymerase chain reaction assay, the transcription levels of miR-29a were higher in the bone tissues harvested from OI mice than from wild-type mice (Figure 2). In situ hybridization observation showed that osteoblasts adjacent to trabecular bone in the OI mice weakly displayed miR-29a transcript compared to the wild-type mice (Figure 3).

We constructed lentivirus-shuttled vectors encoding miR-29a antisense oligonucleotide (miR-29a-AS) and miR-29a precursors (pre-miR-29a) to investigate the role of miR-29a in the bone metabolism of OI mice in the subsequent studies (Figure 4). Mock vector refers to the control lentivirus not suppressing or enhancing the expression of miR-29a. Under the RT-qPCR assay, miR-29a-AS and pre-miR-29a could successfully suppress and enhance the transcription of miR-29a, respectively (Figure 5).

Micro-CT images revealed that the specimens from the OI mice and OI mice treated with mock vector or pre-miR-29a exhibited sparse trabecular and cortical microstructures compared to wild-type mice. Intense trabecular and thick cortical microarchitecture comparable to wild-type mice were visible in the OI-miR-29a-AS-treated mice. In other words, the poor bone microstructure in OI mice could be improved with the miR-29a-AS treatment (Figure 6). As for morphometric parameters, BMD (mg/cm^3^), trabecular bone volume to total volume fraction (BV/TV) (%), and trabecular number (Tb.N) (1/mm) seem to be lowest in OI mice treated with pre-miR-29a, and miR-29a-AS seems to have normalizing effects (Figure 7).

We also assessed the impact of miR-29a-AS and pre-miR-29a on the expression of proliferating cell nuclear antigen (PCNA), β-catenin, a bone-promoting regulator, and Dickkopf-related protein 1 (DKK1), a strong skeletal-deleterious factor by immunohistochemical (IHC) analyses. Compared with the bone tissues from the wild-type mice, the bone tissues from the OI mice showed decreased PCNA, decreased β-catenin, and increased DKK1. The aberrant expression patterns of bone metabolic factors in OI mice could be accentuated by pre-miR-29a and normalized by miR-29a-AS (Figure 8, Figure 9 and Figure 10).

We also assessed the impact of miR-29a-AS and pre-miR-29a on the recruitment of osteoclasts by tartrate-resistant acid phosphatase (TRAP) histochemical staining. Compared with the bone tissues from the wild-type mice, the bone tissues from the OI mice showed increased TRAP staining. The trend of increased osteoclast recruitment, revealed by increased TRAP staining, in OI mice could be deteriorated by pre-miR-29a and normalized by miR-29a-AS (Figure 11).

We isolated primary bone marrow mesenchymal cells (BMSCs) and primary bone marrow osteoclast precursor cells (pre-OCs) from wild-type mice; OI mice; and OI mice treated with mock lentivirus, miR-29a-AS, or pre-miR-29a and verified their ex vivo osteogenic and osteoclastogenic differentiation capacities, respectively. Compared with the BMSCs harvested from the wild-type mice, the BMSCs harvested from the OI mice showed decreased ex vivo osteogenic differentiation capacities under von Kossa staining. The trend of compromised osteogenic differentiation capacity of the BMSCs harvested from OI mice under von Kossa staining could be accentuated by pre-miR-29a and rescued by miR-29a-AS (Figure 12). Compared with the pre-OCs harvested from the wild-type mice, the pre-OCs harvested from the OI mice showed increased ex vivo osteoclastogenic differentiation capacities under fluorescence F-actin staining. The trend of enhanced osteoclastogenic capacity of the pre-OCs harvested from OI mice could be accentuated by pre-miR-29a and rescued by miR-29a-AS (Figure 13).

Finally, we want to investigate whether miR-29a-AS could lead to hepatic or renal injuries. We showed that miR-29a-AS did not significantly change the serum GPT, blood urea nitrogen, or creatinine levels (Figure 14).

## 4. Discussion

Our study showed that the expression levels of miR-29a are higher in bone tissues obtained from the OI mice than from wild-type mice, demonstrated by RT-qPCR and in situ hybridization assay. The inferior bony architecture in micro-CT and pertinent morphometric parameters could be rescued by miR-29a-AS and accentuated by pre-miR-29a. The decreased PCNA, increased DKK1, and decreased β-catenin expression in OI mice could be accentuated by pre-miR-29a and normalized by miR-29a-AS. The decreased osteogenesis and increased osteoclastogenesis in OI mice could also be accentuated by pre-miR-29a and normalized by miR-29a-AS. miR-29a-AS did not seem to possess severe hepatic or renal toxicities.

Several in vitro and ex vivo studies have suggested the role of miRNAs in the regulation of cellular functions of bone cells [10]. Our group has been dedicated to unraveling the role of miR-29a in various bone diseases. MiR-29a protects against glucocorticoid-induced bone loss and fragility in rats by orchestrating bone acquisition and resorption [16]. MiR-29a ameliorates glucocorticoid-induced suppression of osteoblast differentiation by regulating β-catenin acetylation [17]. MiR-29a represses osteoclast formation and protects against osteoporosis by regulating P300/CBP-associated factor (PCAF)-mediated RANKL and C-X-C motif chemokine ligand 12 (CXCL12) [18]. MiR-29a can mitigate osteoblast senescence and counteracts bone loss through oxidation resistance-1 control of forkhead box O3 (FoxO3) methylation [19]. MiR-29a in osteoblasts represses high-fat diet-mediated osteoporosis and body adiposis through targeting leptin [20]. Piezoelectric microvibration mitigates estrogen loss-induced osteoporosis and promotes piezo-type mechanosensitive ion channel component 1 (Piezo1), miR-29a, and Wnt family member 3a (Wnt3a) signaling in osteoblasts [21]. Our present study reported about the role of miR-29a in OI, supplementing the eclectic repertoire of miR-29a in bone diseases.

There is currently no curative remedy for OI [27]. The drugs that are prescribed to human OI patients have been extrapolated from the treatment of osteoporosis. The most used drugs are bisphosphonates inducing the apoptosis of osteoclasts [6]. In Palomo et al.’s study recruited 37 children with OI (OI type I, *n* = 1; OI type III, *n* = 14; and OI type IV, *n* = 22) who initiated intravenous bisphosphonate therapy before 5 years of age (median 2.2 years; range, 0.1 to 4.8 years) and who had a subsequent follow-up period of at least 10 years (median 14.8 years; range, 10.7 to 18.2 years), during which they had received intravenous bisphosphonate treatment for at least 6 years. During the observation period, the mean lumbar spine areal BMD Zeta scores (Z-scores) increased from −6.6 (standard deviation (SD) 3.1) to −3.0 (SD 1.8), and the weighted Z-scores increased from −2.3 (SD 1.5) to −1.7 (SD 1.7) (*p* < 0.001 and *p* = 0.008). At the time of the last assessment, patients with OI type IV had significantly higher height Z-scores than a control group of patients matched for age, gender, and OI type who had not received bisphosphonates. Patients had a median of six femur fractures (range, 0 to 18) and five tibia fractures (range, 0 to 17) during the follow-up period. At baseline, 35% of vertebra were affected by compression fractures, whereas only 6% of vertebra appeared compressed at the last evaluation (*p* < 0.001), indicating vertebral reshaping during growth. Spinal fusion surgery was performed on 16 patients (43%). Among the 21 patients who did not have spinal fusion surgery, 13 had scoliosis, with a curvature ranging from 10 to 56 degrees. In this study, intravenous bisphosphonate therapy was associated with higher Z-scores for lumbar spine areal bone mineral density and vertebral reshaping, but long-bone fracture rates were still high, and the majority of patients developed scoliosis [7]. Bisphosphonate treatment could not effectively mitigate the fracture incidence, and its long half-life (could be several years in some patients) is also a concern [7,27]. In the light of these drawbacks, Denosumab, a monoclonal antibody inhibiting osteoclast activation, has been considered as a promising alternative, since it has a relatively short degradation period defying long-term accumulation. H Hoyer-Kuhn et al. recruited ten patients (age range: 5.0–11.0 years; at least 2 years of prior bisphosphonate treatment) with genetically confirmed OI. Denosumab was administered subcutaneously every 12 weeks at 1 mg/kg body weight. The mean relative change of lumbar BMD was +19 % (95% confidence interval: 7–31%). The lumbar spine aBMD Z-Scores increased from −2.23 ± 2.03 (mean ± SD) to −1.27 ± 2.37 (*p* = 0.0006). The study group reported no severe side effects [8]. However, rebound osteoporosis, hypercalcemia, and hypercalciuria hinder the regular use of Denosumab among OI patients [27]. Due to the shortcomings of antiresorptive agents, further endeavors are directed towards enhancing osteoblasts rather than inhibiting osteoclast function [28]. For instance, teriparatide has been proposed, but it has neither been used nor studied among OI children, due to concerns about potential osteosarcoma observed in animal studies [29].

These clinically unmet needs urged us to search for alternative therapies that could strengthen the bone quality of OI patients. The miRNAs harbor substantial potential to serve as therapeutic agents in bone disorders because of their capacity to regulate the translation of proteins by applying a variety of methods (e.g., degradation of mRNAs or deadenylation of mRNAs) and to overcome the side effects secondary to the administration of drug molecules of interest [11]. The miRNAs can also disturb the function of ribosomes, followed by the destruction of specific nascent polypeptides [11]. Moreover, they could be delivered very easily to the site of infection, a limitation of employing conventional therapeutics [11]. In addition, 39.7% of miRNAs are tissue-specific, making tissue-specific expression possible [30]. The advantages mentioned above make harnessing miRNAs for the treatment of OI a worth investigating revenue. However, the inhibition of miR-29a was achieved by a lentivirus vector, a vector that our lab was highly familiar with, in our study. Insertional mutagenesis is a major concern with lentivirus vectors. As a result, our findings were proof-of-concept in nature only, and the application of a more affable vector for the delivery of miR-29a-inhibiting material is necessary in future studies.

We showed that miR-29a-AS could enhance osteogenesis and suppress osteoclastogenesis, an apparent superiority over bisphosphonate and Denosumab that could only suppress osteoclastnogenesis, and improve the bone architecture rather than just increasing bone mass only.

## 5. Conclusions

Our study showed that the inhibition of miR-29a expression could increase osteogenesis and suppress osteoclastogenesis, optimizing the bone architecture in OI mice (Figure 15). The inhibition of miR-29a activity is an alternative avenue for the treatment of OI warranting further investigation.

## Figures and Tables

**Figure 1 biomedicines-11-00465-f001:**
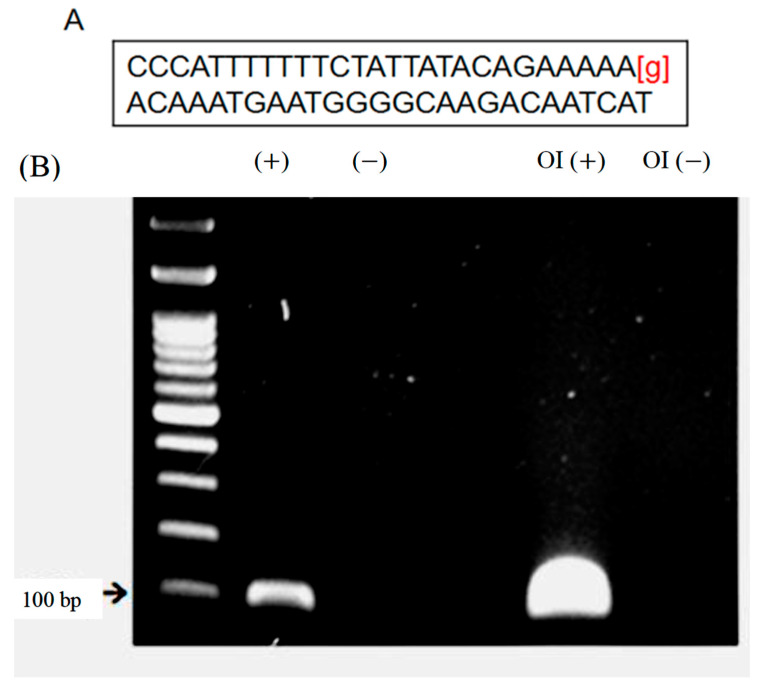
Genotyping of OI mice. (**A**) Manufacturer offered primer sequences for pyrosequencing. (**B**) Representative agarose electrophoresis image of genotyping. (+): positive control; (−): negative control; OI (+): genomic DNA from OI mice; OI (−): genomic DNA from non-OI mice.

**Figure 2 biomedicines-11-00465-f002:**
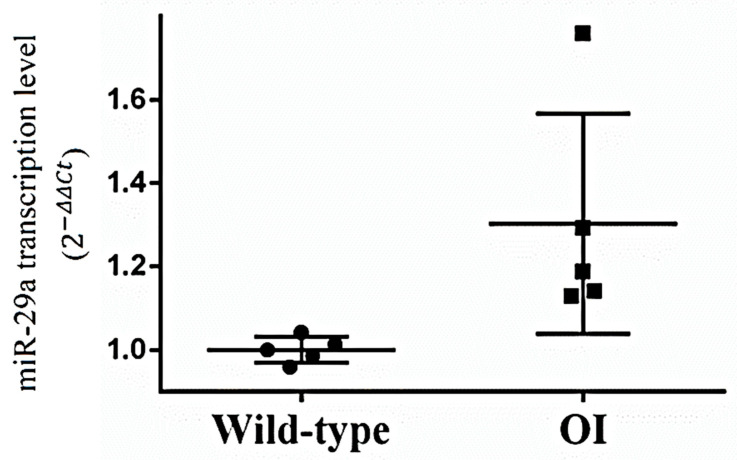
The transcription levels of miR-29a for the bone tissues harvested from wild-type mice and OI mice (*n* = 5 for each group).

**Figure 3 biomedicines-11-00465-f003:**
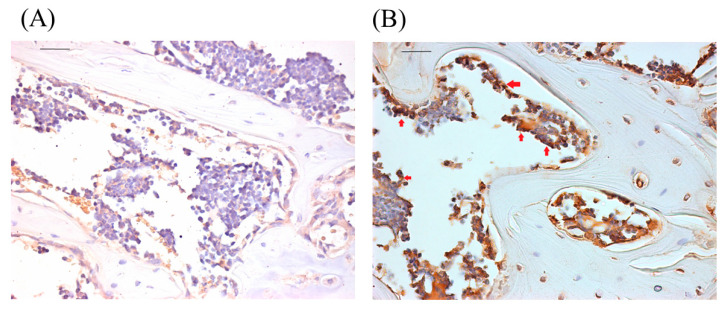
Representative in situ hybridization images targeting miR-29a for the trabecular bone tissues from wild-type (**A**) and OI mice (**B**). Scale bar represents 100 μm. The red arrows indicate the regions expressing miR-29a.

**Figure 4 biomedicines-11-00465-f004:**
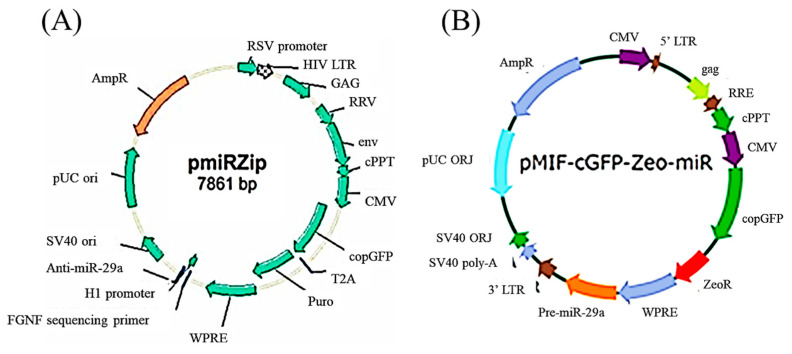
Gene maps of lentivirus-shuttled (**A**) miR-29a-AS and (**B**) pre-miR-29a.

**Figure 5 biomedicines-11-00465-f005:**
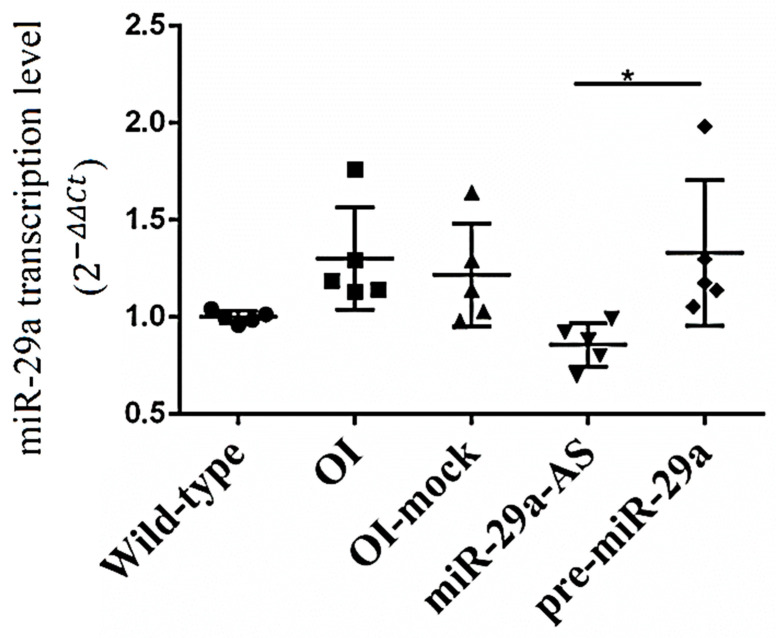
The transcription levels of miR-29a for wild type mice, OI mice and OI mice treated with mock, miR-29a-AS, and pre-miR-29a vectors (*n* = 5 for each group) (* *p* < 0.05).

**Figure 6 biomedicines-11-00465-f006:**
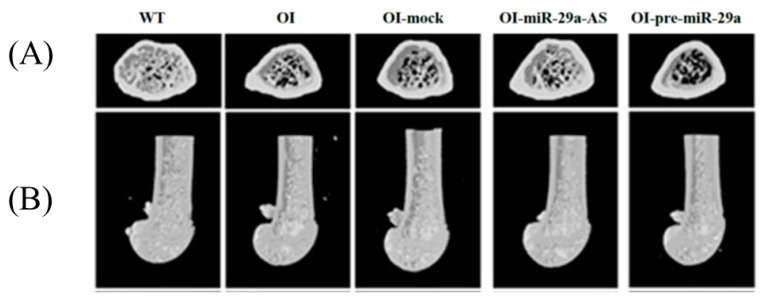
Representative micro-CT images of bony microstructures in (**A**) lumbar spine and (**B**) distal femur.

**Figure 7 biomedicines-11-00465-f007:**
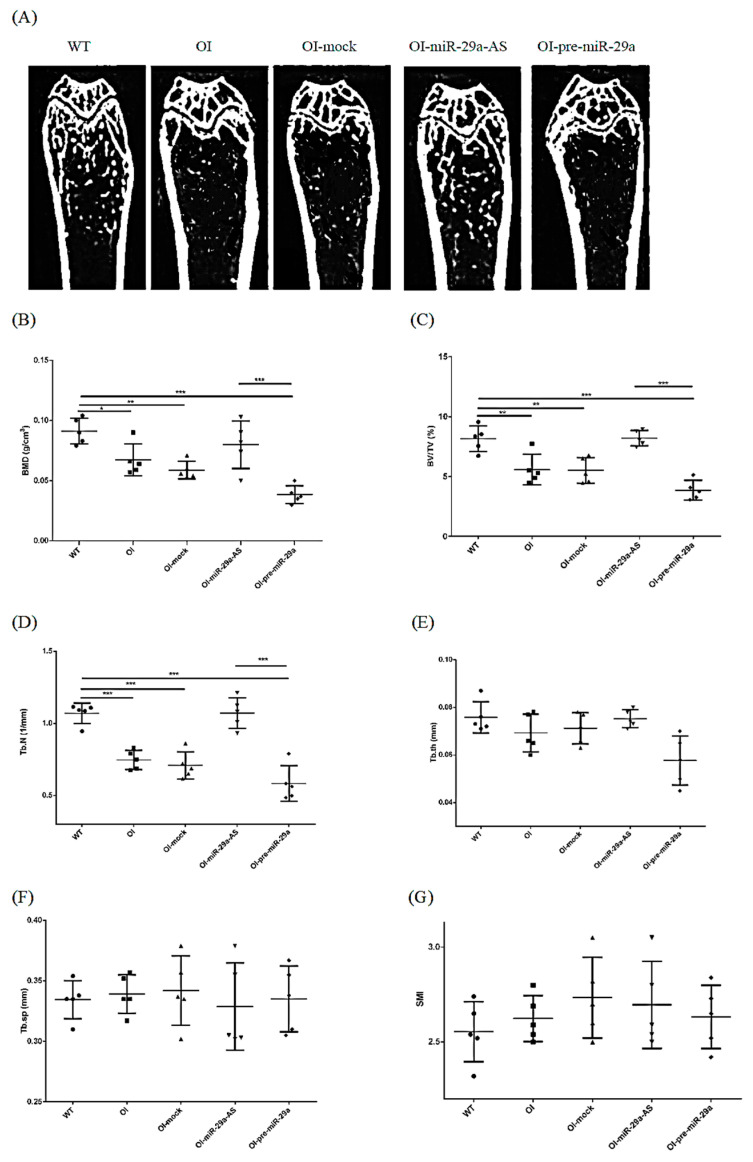
The representative micro-CT images and microstructure parameters. (**A**) The representative micro-CT images of the distal femur. (**B**) BMD: bone mineral density; (**C**) BV/TV: trabecular bone volume to total bone volume fraction; (**D**) Tb.N: trabecular number; (**E**) Tb.th: trabecular thickness; (**F**) Tb.sp: trabecular separation; (**G**) SMI: structure model index (*n* = 5 for each group) (* *p* < 0.05, ** *p* < 0.01, and *** *p* < 0.001).

**Figure 8 biomedicines-11-00465-f008:**
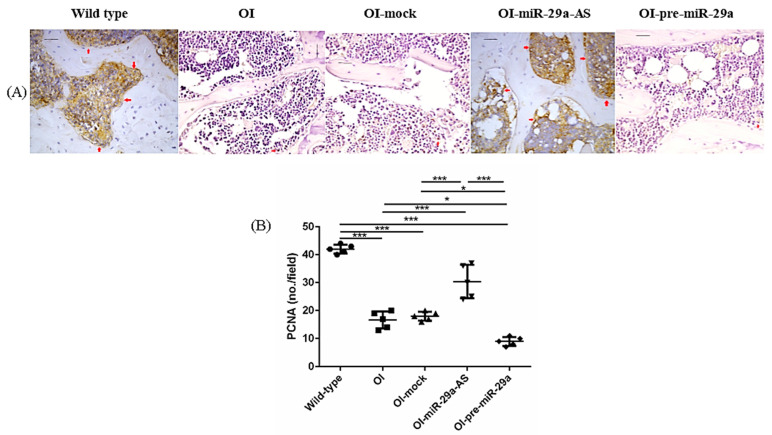
Immunohistochemical images of proliferating cell nuclear antigen (PCNA) in bone tissue (**A**) and quantification figure (**B**). Red arrows indicated the positively stained region (* *p* < 0.05 and *** *p* < 0.001) (*n* = 5 for each group). Scale bar represents 100 μm.

**Figure 9 biomedicines-11-00465-f009:**
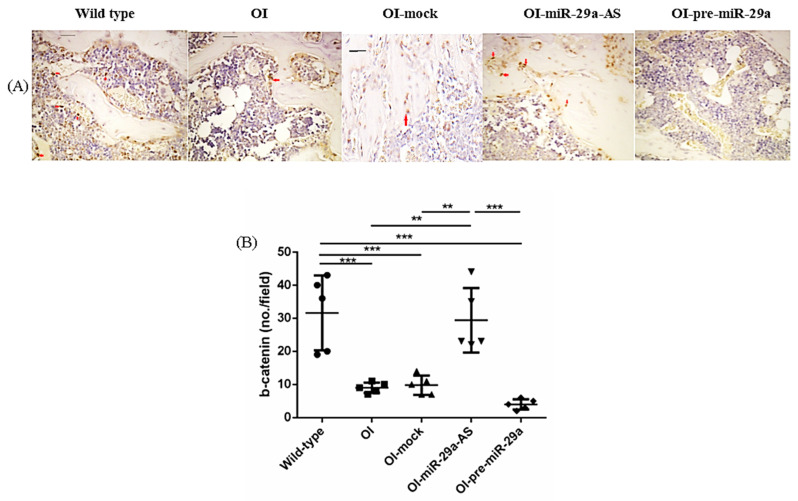
Immunohistochemical images of β-catenin in bone tissue (**A**) and quantification figure (**B**). Red arrows indicated the positively stained region (** *p* < 0.01 and *** *p* < 0.001) (*n* = 5 for each group). Scale bar represents 100 μm.

**Figure 10 biomedicines-11-00465-f010:**
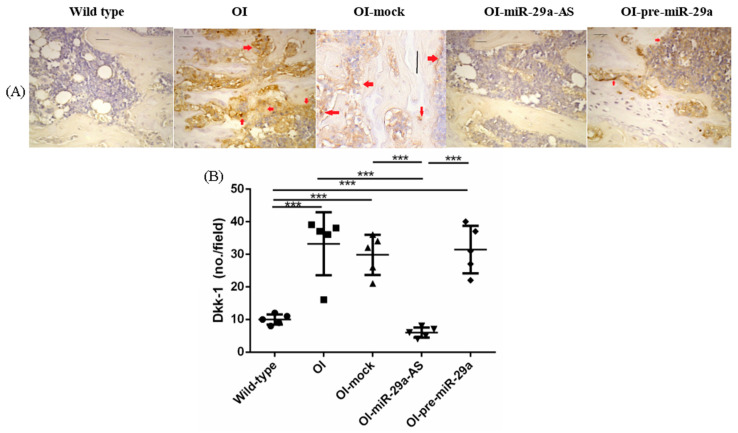
Immunohistochemical images of Dickkopf-related protein 1 (DKK1) in bone tissue (**A**) and quantification figure (**B**). Red arrows indicated the positively stained region (*** *p* < 0.001) (*n* = 5 for each group). Scale bar represents 100 μm.

**Figure 11 biomedicines-11-00465-f011:**
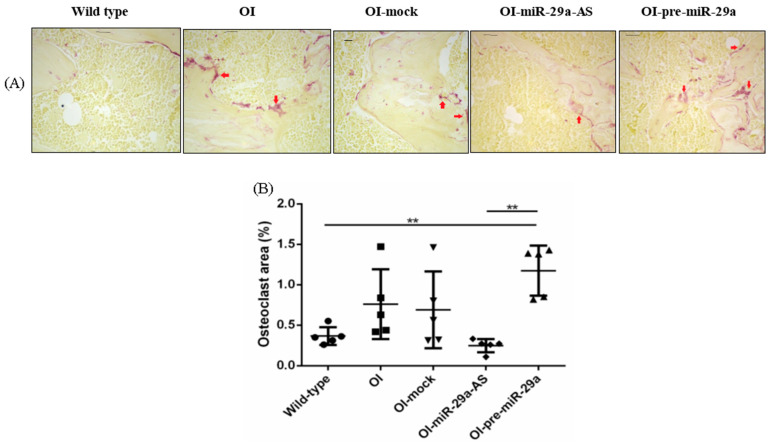
Immunohistochemical images of tartrate resistant acid phosphatase (TRAP) in bone tissue (**A**) and quantification figure (**B**). (** *p* < 0.01) (*n* = 5 for each group).

**Figure 12 biomedicines-11-00465-f012:**
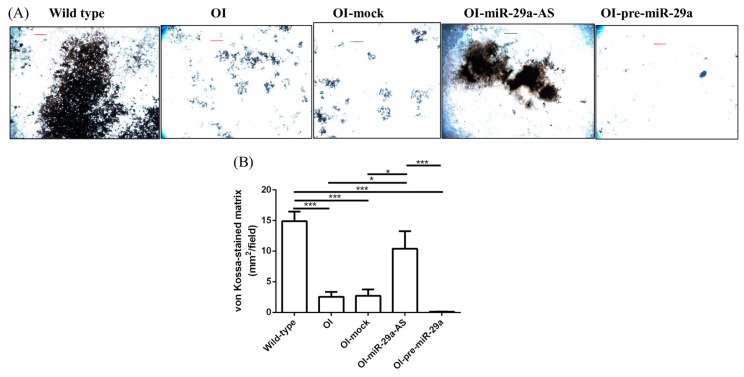
**The capacity of ex vivo osteogenic differentiation of bone marrow mesenchymal cells**. (**A**) Illustration of mineralized nodules by von Kossa cytochemical staining. (**B**) The area of von Kossa-stained matrix under osteogenicconditions (*n* = 5 for each group) (* *p* < 0.05 and *** *p* < 0.001). Red scale bar indicated 2 mm.

**Figure 13 biomedicines-11-00465-f013:**
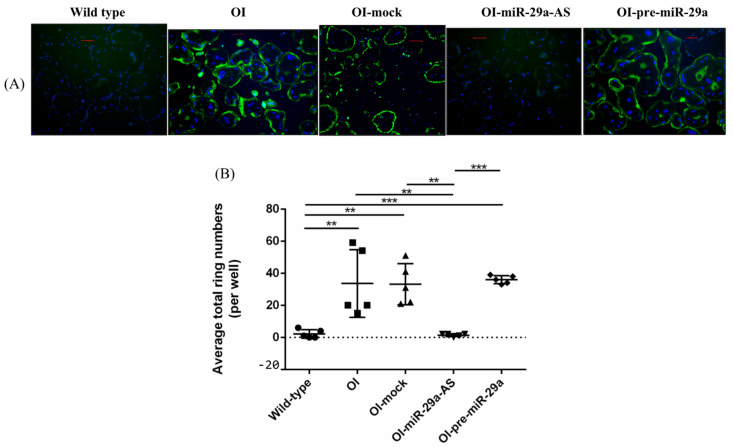
The capacity of ex vivo osteoclastogenic differentiation of bone marrow osteoclast precursor cells. (**A**) Illustration of fluorescence F-actin ring formation. (**B**) The number of F-actin ring formation under osteoclastogenic conditions. The total number of F-actin rings in 3 different fields (×100 magnification) per well, and 3 wells per animals were counted (** *p* < 0.01 and *** *p* < 0.001) (*n* = 5 for each group). Scale bar represents 200 μm.

**Figure 14 biomedicines-11-00465-f014:**
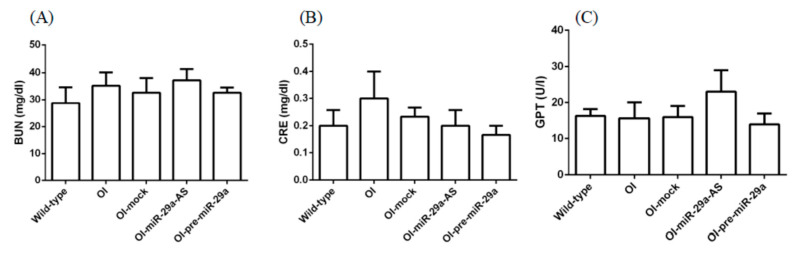
The serum levels of BUN (**A**), creatinine (**B**), and GPT (**C**) among the five groups (*n* = 5).

**Figure 15 biomedicines-11-00465-f015:**
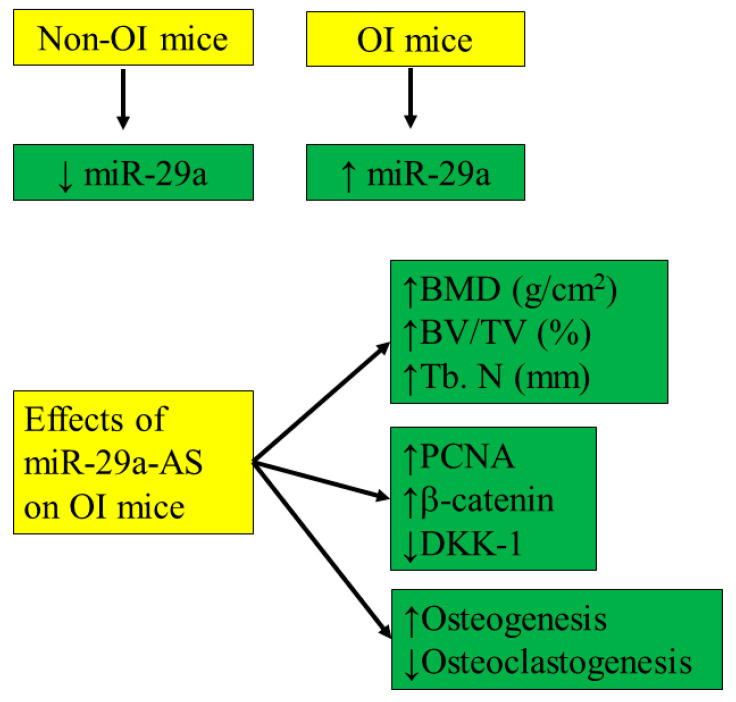
The schematic summary of the current study.

## Data Availability

The data presented in this study are available on request from the corresponding author.

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
