# Peer review of "Micro Ribonucleic Acid−29a (miR−29a) Antagonist Normalizes Bone Metabolism in Osteogenesis Imperfecta (OI) Mice Model"

_biomedicines, 2023, doi:10.3390/biomedicines11020465_

Round 1

Reviewer 1 Report

This manuscript by Jih-Yang Ko et al investigated the role of miRNA 29a in OI development and its potential by regulating miRNA29a using antisense oligonucleotide or pre-miRNA 29a and found miRNA 29a is upregulated in OI model and systemic knocking down miRNA can rescue the bone deterioration phenotype. The study is based on several previous publications on miRNA29a. However, their previous publication showed that forced miRNA 29a promotes osteogenesis and inhibits adipogenesis.  The new finding in this study appears to contradict the previous study (PMID:34502056) despite using different animal models. Overall, the authors provided a lot of data. But the quality of presented data was comprised by their way of presenting without details. There are quite a few grammar errors. Therefore, English writing needs to be improved. 

Following are my comments:

Title: “Antagonism” should be “Antagonist”.

Abstract: Line 15, same issue as title.

Line 16, “In higher” should be is or was higher”?

Line 17, “Under RT-Q-PCR” Under should be replaced with an alternative word such as “demonstrated” or “showed”.

Line 20, “Accentuated” maybe replaced with deteriorated

Line 60: “We want to” may be replaced with” We aim to” to be more professional.

Line 64, “In our study” is not specific here, it should be: “In this study or in the current study”.

Line 82, “What is miRNA 29a inhibitor? Does the author mean antisense oligonucleotide?

Methods section:

Methods should be written in a logical way, it is better to break them down into subsections using subheadings.

Such as microCT, Histology, RT-PCR, in vitro osteogenesis, in vitro osteoclastogenesis, statistical analysis and so on.  For RT-PCR, RNA extraction first, then reverse transcription, then Q-PCR.

Results: Results need to be written in a logical way.

1.      Figure 1, what is the difference of (+) and OI(+) and (-) and OI(-)? Figure legends should add some explanation to these labels.

2.      Figure 2, was miRNA-29a expression in OI mice significantly higher than normal mice, from the graph, it only increase about 30%.  Why the control group has the same value, did the control has no variabilities? Probably the authors did not calculate 2delta-Delta CT  correctly. Both control and experiment groups should subtract the average Delta-CT of control group so that both groups have error bars or SD when converted to fold changes.

3.      Figure 3, What parts of the bone tissues are the authors showing? Those parts presented do not look like trabecular bone or cortical bone. I cannot tell which cells express miRNA29a. The figure legend should state which gene in situ hybridization was performed.

4.      Figure 6, There is no A or B in the figure panel while the figure legend showed A or B. Apparently the top panel is the spine while the bottom two panels are the femurs. Authors should state femur instead of long bone to be more specific.

5.      Figure 7, bone parameters are opposite to the claim that miRNA29a play negative role because this group of mice showed the lowest BMD, BV/TV, Tb.N  and relative high Tb.Sp. These results do not match micro-CT images. Authors should also state which part of the trabecular bone these parameters are generated from. So, it is best to show the micro-CT images and microstructure parameters in the same figure.

6.      Figure 8. Immunohistochemistry images are blurry, insets highlighted positive cells and the location of these cells should be used. Magnification or scale bars should also be provided. What marker is stained such as PCNA, beta-catenin, DKK-1 should be placed on left side of the staining panel to be more straightforward so that readers can know which staining it is without the need to look at the figure legends. Quantification graphs should also have a letter label.  Is PCNA proliferating cells completely devoid in OI, OI+ mock or OI+pre-miRNA29a? They are so clear like negative control. I believe bone marrow definitely has PCNA + cells if osteoblasts or osteoclasts do not proliferate in OI. OI+mock appeared to express high beta-catenin. Quantification do not match images.

7.      Line 226, “Evidence by” should be “revealed by”. Evidence is a noun.

8.      Line 227, “accentuated” better replaced with “deteriorated”. Many other places they also used this word and need to be corrected.  

9.      Figure 9 should use A, B,C panels to be clear for images and quantification. TRAP staining images are blurry. Counterstain nuclei are not visible. The authors said the trend of decrease of TRAP in the graph, but not showing statistical significance, therefore exact P value should be provided.

10.  Figure 10 images of Von Kossa staining should have magnification and scale bars. The nuclear counterstain was not performed. The quantification histogram is blurry. A,B,C and so on panel labels should be given. Figure legends, * should be indicated.

11.  Figure 11, DAPI staining is only evident in OI group.

Discussion: The authors spend a lot effort (half page) in criticizing current therapies for osteoporosis while only the last paragraph discusses miRNA29a they investigated in the current study. Although miRNA19a appeared to promote osteogenesis and inhibit osteoclastogenesis, they need to be delivered using a lentiviral vector and safety is a concern. So, it is better not to oversell their results.  The discussion needs to be revised significantly by focusing more miRNA 29a as a potential therapy for OI.

Line 265, “Under” should be replaced with “as demonstrated or showed”.

Line 272: last sentence should be removed.

Line 290 “Children 37 Children” What this mean?

Author Response

Please see the attached file, thanks!

Reviewer 2 Report

Ko et co. presents the researches performed to evaluate the influence of microRNA-29a antagonism in an experimental model of osteogenesis imperfecta in mice.

Various changes are strongly suggested prior to re-submission of the manuscript at the present journal.

The introduction section should be more complete, providing supplementary background in the field.

The results obtained should be compared with those achieved by other researchers and discussions should be significantly detailed.

(see:

·   Bravo Vázquez LA et al. The Emerging Role of MicroRNAs in Bone Diseases and Their Therapeutic Potential. Molecules 2021;27(1):211.

·   Lehmann TP et al. The regulation of collagen processing by miRNAs in disease and possible implications for bone turnover. Int J Mol Sci 2022; 23: 91.

·   Sharma AR et al. Recent advancements of miRNAs in the treatment of bone diseases and their delivery potential. CRPHAR 2023; 4: 100150.

and others)

In result and discussion section, the authors need to develop argumentation in depth based on the current understanding and the findings of the results obtained, presenting the potential, the weakness and limitation, and future research direction, among others. Authors should try to explain the theoretical implication as well as the translational application of their research.

Some other aspects were found in this manuscript:

- missing information about the conditions of housing the laboratory animals;

- many of abbreviations are not explained (see: RT-qPCR, Micro-CT, RNA, DNA, CsCl, PBS, α-MEM, DMEM, RANKL, CXCL 12, Piezo1, FoxO3, SD, 95%CI, Z-score).

- all abbreviations should be expanded in the first appearance and should not be repeated in order to decongest the text and facilitate the understanding of the information transmitted.

- in the text, two terms, which actually represent the same thing (microRNA-29a and miRNA-29a), are used alternately, creating confusion.

- different fonts were used in the text and in the figures;

- missing information (manufacturer, or city, or country) about the companies producing some devices or kits used in the researches (lines: 68, 73, 81, 85, 90, 104, 105, 110, 117, 124, 126, 145 and others);

- the paragraph from lines 280-285 should be reformatted;

- the references should be upgraded;

- the references should be described according to the journal`s guide, using the abbreviated journal name, the journals issues, pages (missing the journal`s name; see: references 1, 2, 3, 4, 6, 9, 10, 19, 20, 21, 23, 24, 26, 30);

- at the references the authors should provide the DOI of the all the articles

- a schematic representation of the study would be appreciated;

- spelling check of the text is mandatory;

- English including grammar, style and syntax, should be improved through the professional help from English Editing Company for Scientific Writings.

Author Response

Please see attached file, thanks !

Round 2

Reviewer 1 Report

The authors have addressed all my comments. Congratulations, the manuscript is acceptable for publication in Biomedicine! 

Reviewer 2 Report

The authors mostly responded to the comments and suggestions and the manuscript was revised accordingly. I consider it could be accepted for publication in this journal, but I propose to have the manuscript checked by a native English speaking person.